# Optimized Deep Brain Stimulation Surgery to Avoid Vascular Damage: A Single-Center Retrospective Analysis of Path Planning for Various Deep Targets by MRI Image Fusion

**DOI:** 10.3390/brainsci12080967

**Published:** 2022-07-22

**Authors:** Xin Wang, Nan Li, Jiaming Li, Huijuan Kou, Jing Wang, Jiangpeng Jing, Mingming Su, Yang Li, Liang Qu, Xuelian Wang

**Affiliations:** 1Department of Neurosurgery, Tangdu Hospital, The Fourth Military Medical University, Xi’an 710038, China; wx2002@163.com (X.W.); linan314@fmmu.edu.cn (N.L.); doc_leejm@163.com (J.L.); wmwangj@163.com (J.W.); drjingjp@xjtufh.edu.cn (J.J.); summgs163-ok@163.com (M.S.); sdbzliyang@126.com (Y.L.); poplol909@163.com (L.Q.); 2Department of Cardiology, The Second Affiliated Hospital, Xi’an Jiaotong University, Xi’an 710004, China; khjsarah@163.com

**Keywords:** co-registration, deep brain stimulation, magnetic resonance imaging, intracranial hemorrhage, trajectory planning

## Abstract

Co-registration of stereotactic and preoperative magnetic resonance imaging (MRI) images can serve as an alternative for trajectory planning. However, the role of this strategy has not yet been proven by any control studies, and the trajectories of commonly used targets have not been systematically studied. The purpose of this study was to analyze the trajectories for various targets, and to assess the role of trajectories realized on fused images in preventing intracranial hemorrhage (ICH). Data from 1019 patients who underwent electrode placement for deep brain stimulation were acquired. Electrode trajectories were not planned for 396 patients, whereas trajectories were planned for 623 patients. Preoperative various MRI sequences and frame-placed MRI images were fused for trajectory planning. The patients’ clinical characteristics, the stereotactic systems, intracranial hemorrhage cases, and trajectory angles were recorded and analyzed. No statistically significant differences in the proportions of male patients, patients receiving local anesthesia, and diseases or target distributions (*p* > 0.05) were found between the trajectory planning group and the non-trajectory planning group, but statistically significant differences were observed in the numbers of both patients and leads associated with symptomatic ICH (*p* < 0.05). Regarding the ring and arc angle values, statistically significant differences were found among various target groups (*p* < 0.05). The anatomic structures through which leads passed were found to be diverse. Trajectory planning based on MRI fusion is a safe technique for lead placement. The electrode for each given target has its own relatively constant trajectory.

## 1. Introduction

The key surgical procedure for deep brain stimulation (DBS) is electrode insertion into the cerebral nuclei using stereotactic techniques. DBS is an important technique in the treatment of functional brain disorders and ameliorates neurological and psychiatric symptoms in patients. This minimally invasive technique has a lower risk of intracranial hemorrhage (ICH) than craniotomy, but ICH may still occur.

A potentially effective method to reduce ICH caused by lead implantation surgery is to plan the trajectory of implanted electrodes using a variety of stereotactic surgical planning software programs to avoid ventricles, sulci, and blood vessels in the brain [1,2]. However, the incidence of ICH before and after the implementation of neuronavigation has been reported to be the same [3]. Although images with contrast enhancement were utilized to facilitate trajectory planning to avoid vessel injury, one study reported a relatively high rate of postoperative ICH (13/272) [2]. Careful trajectory planning tends to be helpful to prevent hemorrhage but cannot fully prevent venous hemorrhagic infarctions, which may cause more serious neurological sequelae [4,5]. Recent studies on trajectory planning for DBS revealed different incidence rates of ICH, which varied from 0.7 to 2.5% per electrode [6,7]. Even the notion that transventricular trajectories significantly increase the risk of neurologic complications has been challenged because only one confirmed case of postoperative intraventricular asymptomatic hemorrhage was noted among 206 patients [8]. Taken together, trajectory planning may facilitate ICH prevention, but reliable case–control studies and convincing evidence are needed to verify this assumption. Since 2015, our center has adopted image fusion technology to automatically fuse preoperative magnetic resonance imaging (MRI) data from three-dimensional brain volume (3D BRAVO) imaging, susceptibility-weighted imaging (SWI), time-of-flight magnetic resonance angiography (TOF MRA), T1-weighted gadolinium-enhanced MRI (T1W-Gd), and MRI performed after stereotactic frame placement, and the combined data have been used to plan a path for lead implantation (Appendix A). To further confirm the role of trajectory planning based on multiple merged images in preventing vascular complications, we performed this retrospective study.

Furthermore, a trajectory passing through the caudate nucleus may affect Parkinson’s disease (PD) patients’ cognitive flexibility and attentional–executive functions after subthalamic nucleus (STN) DBS surgery [9,10]. The microlesion of white matter bundles associated with STN electrode trajectories may lead to the worsening of PD patients’ verbal fluency [11]. These findings reflect the necessity of studying neuroanatomical descriptions of the brain along the trajectories of electrodes implanted at frequently used targets. Our study provides trajectory angles and the brain structures intersected for lead placement at the STN, ventralis intermedius nucleus of the thalamus (Vim), globus pallidus internus (GPi), and nucleus accumbens (NAc) for the first time.

## 2. Materials and Methods

### 2.1. Patient Demographics and Clinical Characteristics

In this study, conducted at our hospital between January 2009 and December 2019, 1019 patients with a total of 1928 implanted electrodes underwent DBS. The lead implantation path was not planned for 396 patients who underwent implantation with a total of 691 electrodes before December 2014, whereas stereotactic surgical planning software was used to plan the electrode trajectories for 623 patients who received a total of 1237 implanted electrodes after January 2015. This study was performed in accordance with the Declaration of Helsinki and approved by the Institutional Review Board of Tangdu Hospital, the Fourth Military Medical University. The ethical code number was TDLL-201706-28. All patients and appointed agents signed operation agreements and related documents.

### 2.2. Stereotactic Systems and Implanted Devices

We used the Leksell (Elekta AB, Stockholm, Sweden) or CRW (Radionics Inc., Burlington, MA, USA) stereotactic systems and accompanying anatomical targeting software to fuse images and plan trajectories. The stereotactic systems and their corresponding software programs were as follows: Leksell—SurgiPlan/ImageMerge/ImagefusionTM, CRW—StereoCalcTM/NeuroSight Arc/ImageFusionTM, Leksell and CRW—StealthStation S7 Framelink 5 (Medtronic, Inc., Minneapolis, MN, USA). The models of the implantable electrodes were as follows: 3389/3387/3391-40 or 40s (Medtronic, Inc., USA), L301/302 (Beijing Pins Medical Equipment Co., Ltd., Beijing, China), and 1200/1210 (SceneRay Co., Ltd., Suzhou, China).

### 2.3. Surgical Procedure and Trajectory Planning

The lateral beam of the coordinate frame must be maintained parallel to the line connecting the nasal ala and earlobe (the anterior commissure (AC)–posterior commissure (PC) surface projection line) when positioning the stereotactic frame. The patients in whose implantation surgery the lead paths were not planned underwent 1.5 T or 3.0 T MRI scans of the brain regions surrounding the targets on the day of the operation. T_1- or T_2-weighted (T1W or T2W, respectively) imaging (2- or 3-mm slice thickness, 0-mm slice gap) was selected according to the nucleus. For instance, T2W imaging was selected for the STN, and T1W imaging was selected for the other nuclei. Stereotactic surgical planning software was used to locate the targets directly and record their coordinates. Bilateral burr holes were often located 0.5~1.0 cm anterior to the coronal suture and 3.5~4.0 cm lateral to the midline. Except for minimal adjustment of the entry point in the same burr hole in the presence of a visible superficial brain vein, no other means were utilized to plan and optimize the trajectories. Accordingly, this surgical procedure can also be called “hole-based trajectory”.

The patients in whose implantation surgery the lead paths were planned underwent 1.5 T or 3.0 T MRI scans, including 3D BRAVO imaging, a few days before surgery. Some of these patients additionally underwent MRI scans such as SWI, TOF MRA, and T1W-Gd. The technical details of the MRI scans are presented in Appendix A. On the day of the operation, each patient underwent 1.5 T or 3.0 T MRI scans of the brain regions surrounding the targets after the stereotactic frame was placed. An orthogonal radio frequency (RF) coil was used for head. The scanning parameters were identical to those for the group in which trajectory planning was not used. We applied the corresponding software to merge the images from preoperative MRI and MRI performed after frame placement. After completing targeting on a sequence of images in which the nucleus was more visible, we confirmed the entry points on the scalp and skull against sagittal 3D BRAVO images, which were usually less than 1.5 cm anterior to the coronal suture. Subsequently, a trajectory for cannula puncture and electrode placement was planned to avoid the sulci and ventricles. During the process, alternating between 3D BRAVO imaging and the SWI, TOF MRA, or T1W-Gd data set was important to observe whether the trajectory passed through arteries and veins. In most cases, the trajectory was maintained at least 4 mm from all sulci, ventricles, and blood vessels. Trajectory planning was adjusted if necessary. Finally, two angle values on each side of the skull were recorded. Considering the above process, this surgical procedure can be referred to as “imaging-based trajectory”.

Both in the “hole-based trajectory” and “imaging-based trajectory” surgical procedures, the trajectory was defined by the center of arc principle, and the target points, namely the center of arc, were defined by the neurosurgeon in the surgery planning software based on the MRI images, which had been imported in advance. All the nuclei were targeted directly; the AC-PC line only helped us to confirm the coordinates of nuclei through the relatively constant spatial relationship between them. The differences between these two procedures were that the relatively standardized burr hole and the target point determined the path in the former procedure, and the well-planned trajectory and the target point determined the burr hole in the latter procedure.

Single or bilateral, straight or semicircular frontal incisions were made. In the trajectory planning group, the stereotactic arcs were adjusted according to the two angle values to create marks on the scalp and skull before devising incisions and drilling holes, respectively. Thus, electrode implantation could be guided accurately. The patients required local or general anesthesia for electrode implantation. During the same surgical procedure or a second procedure within 1 week postoperatively, an implantable pulse generator (IPG) was implanted and connected under general anesthesia.

### 2.4. Angle Recording

Two angle values for each target were used to help the surgeons to adjust the stereotactic arc to ensure the accuracy and safety of implantation. The ring and arc angle values computed by Leksell SurgiPlan software (Version 10.1, Elekta AB, Stockholm, Sweden) were recorded for this study (Appendix A).

### 2.5. Diagnosis of ICH

Changes in patients’ vital signs and consciousness after DBS surgery were carefully observed. In the early stage of this study, patients who showed related symptoms and signs such as coma, epileptic seizure, severe headache, and vomiting underwent cranial computed tomography (CT) at the onset of the signs and were then diagnosed with symptomatic ICH. In the later stage of the study, a routine cranial CT scan was performed on the first day after each patient’s operation in the trajectory planning group. If the patient’s condition worsened, an emergency CT scan was required. In addition to symptomatic ICH, most cases of asymptomatic ICH could be diagnosed promptly when the routine CT was used. The recruited patients had a regular follow-up for programming, allowing all late-onset symptomatic ICH events and other severe side effects to be recorded.

### 2.6. Statistical Analysis

Statistical analyses were performed with the Statistical Package for the Social Sciences for Windows (SPSS, version 22.0, IBM, Armonk, NY, USA). The median and the first and third quartiles were used to describe age at surgery. Pearson’s chi-squared test, Fisher’s exact test, and the Mann–Whitney U test were used to compare clinical characteristics between the no trajectory planning group and the trajectory planning group. Pearson’s chi-squared test was used to compare the surgery planning systems used, and Pearson’s chi-squared test and Fisher’s exact test were used to compare the occurrence of ICH between the two groups. The mean ± standard deviation (SD) was used to describe angle values. The Kolmogorov–Smirnov test and Shapiro–Wilk test were used to examine whether the values followed a normal distribution. Bartlett’s test and Levene’s test were used to test the homogeneity of variance. Analysis of variance (ANOVA) and the least significant difference (LSD) test were used to compare ring and arc angle values of trajectories among various targets in the trajectory planning group. For all tests, *p* < 0.05 was considered statistically significant.

## 3. Results

### 3.1. Comparison of Clinical Characteristics between the Two Groups

Aggregate data from all the patients regarding gender, age at surgery, diagnosis, the prevalence of hypertension, and the targets of DBS are shown in Table 1. No statistically significant differences in the proportion of male patients, patients receiving local anesthesia, or the distribution of diseases or targets were found between the trajectory planning group and the group with no trajectory planning (*p* > 0.05). Significant differences in the proportion of patients with hypertension and age at surgery were identified between the two groups (*p* < 0.05).

### 3.2. Comparison of the Stereotactic Systems Used between the Two Groups

A statistically significant difference in the commercial software applied for stereotactic surgery was found between the trajectory planning group and the non-trajectory planning group (*p* < 0.05), but the significant difference in the stereotactic headframe was not found (*p* > 0.05). The stereotactic systems and surgery planning software used in this study are shown in Table 2.

### 3.3. Comparison of ICH after DBS Surgery between the Two Groups

Statistically significant differences in the numbers of both patients and leads associated with symptomatic ICH were observed between the trajectory planning group and the non-trajectory planning group (*p* < 0.05). No symptomatic ICH patients had microelectrode recordings (MER), which were seldom used in this series. All bleeding cases were PD patients with the STN as their targets. The details of the patients with symptomatic ICH in the non-trajectory planning group were published in our previous article [12]. In the trajectory planning group, two patients developed asymptomatic ICH as observed on routine postoperative CT images. In the non-trajectory planning group, regular postoperative imaging was not performed in 186 patients, whose cranial CT scan was not performed on the first day after operation. The symptomatic ICH cases after DBS surgery in the two groups are shown in Table 3. CT images of a PD patient with symptomatic ICH after DBS surgery in the non-trajectory planning group are shown in Figure 1.

### 3.4. Image Fusion of Various MRI Sequences in the Trajectory Planning Group

Various MRI sequences were fused by commercial software to help us to plan a lead path before DBS surgery. Table 4 shows the six combinations used.

### 3.5. Trajectory Angles Calculated by SurgiPlan for Four Targets

No statistically significant differences in ring angle values were found between the STN group and the Vim group, between the STN group and the GPi group, and between the Vim group and the GPi group (*p* > 0.05), but statistically significant differences were identified in all other comparisons between the groups (*p* < 0.05). No statistically significant differences in arc angle values were observed between the STN group and the Vim group (*p* > 0.05), but statistically significant differences were identified in all other comparisons between groups (*p* < 0.05). The ring and arc angle values of the trajectories for four targets are shown in Figure 2.

Additionally, we recorded the ring and arc angle values, which were automatically calculated by the Leksell SurgiPlan software in the trajectory planning group. The incidence of ICH in this group was very low, and all the three symptomatic hemorrhage patients were diagnosed with PD and received the STN-DBS surgery. In spite of the small sample size of symptomatic ICH patients, whose hemorrhages all occurred in the left hemisphere, we preliminarily compared the left ring and arc values between the ICH cases and no ICH cases in the trajectory planning group to define the “ring and arc zone of danger” for lead placement in the STN. However, significant differences were not found (*p* > 0.05). The comparisons of ring and arc angle values of the trajectories between the no ICH cases and ICH cases are shown in Appendix A.

### 3.6. Images of Brain Regions in the Planned Paths of Electrodes for Four Targets

At the levels of the upper part of the lateral ventricle and the internal capsule, the trajectories planned for electrode placement in the four nuclei passed through different anatomic structures in the brain. Simulation of a trajectory from the entry point to the target is shown in Figure 3 and Figure 4.

The planned trajectory for the STN passes posterolaterally to the lateral ventricle close to its lateral wall and then through the posterior limb of the internal capsule (PLIC) to the target. The planned trajectory for the Vim passes posterolaterally to the lateral ventricle close to its lateral wall and then medially to the PLIC to enter the thalamus from its dorsolateral part. The trajectory for the GPi passes laterally to the lateral ventricle far away from its lateral wall and then laterally to the PLIC to enter the globus pallidus from its dorsal part. The trajectory for the NAc passes anterolaterally to the lateral ventricle near its lateral wall and then through the anterior limb of the internal capsule (ALIC) to the target.

Figure 5 shows a rare non-hemorrhagic edematous lesion around the trajectory in the trajectory planning group. An electrode placed for the left STN through the PLIC led to local edema of the PLIC and adjacent white matter (i.e., the medial medullary lamina), which appeared similar to a “hamburger”, together with the relatively normal GPi. These images were taken from a 71-year-old male patient with PD 3 days after he received bilateral lead placement in the STN. He suffered from transient urinary incontinence and mild weakness of the right limbs after the DBS operation.

### 3.7. Avoiding Cerebral Vessels Using Various MRI Sequences

Most of the electrodes were unintentionally kept away from large intracranial arteries in both the hole-based trajectory and imaging-based trajectory cases for commonly used nuclei. A simulated trajectory for the STN and the cerebral arteries on 3D MRA are shown in Figure 6. Moreover, the small veins in the subcortex and the arteries in the sulci and fissures were easily bypassed at close range by the cannula. Some planned trajectories and their surrounding vessels in various MRI sequences are shown in Figure 7.

## 4. Discussion

DBS surgery is associated with a risk of a hemorrhagic complication. The primary task faced by scientists and engineers is the development of a safe, effective, and ideally noninvasive technique to address this problem. The main challenges for neurosurgeons, however, are the awareness and prevention of ICH.

The predisposing factors for ICH include age, gender, perioperative hypertension, and the use of anticoagulants [13,14]. The probability of hemorrhage caused by hypertension may be as high as 10.7% [15] and is higher in patients older than 60 years [16]. In our study, no patients were on a preoperative anticoagulant or antiplatelet drug regime. Their coagulation profiles and platelet counts were all within normal limits. Any patient with significantly abnormal results was excluded from DBS surgery. Underlying diagnoses and brain targets may also be correlated with the risk of ICH. Pediatric dystonia patients had an increased incidence of ICH [17]. The incidence of ICH has been reported to be 2.5% per lead for the STN, 6.7% for the GPi, and 0% for the Vim [18]. According to the study results, no significant differences in the proportion of male patients, disease types, or brain targets were found between the two groups, but significantly more hypertensive and elderly patients were noted in the trajectory planning group than in the no trajectory planning group. Considering that hypertension and advanced age are risk factors for ICH, the data are still believed to be comparable between the two groups and more convincingly attribute decreased hemorrhage events to trajectory planning. In addition to the above factors, the surgical procedure is considered to be the most relevant factor for ICH. Increased safety is associated with a combination of routine general anesthesia, targeting guided only by MRI, and a single-penetration implantation technique [19]. Furthermore, the hemorrhage risk possibly increases as the number of microelectrode passages increases [20]. We reviewed our case series carefully and found that MER was utilized to refine targeting in 31 patients in the no trajectory planning group, and used for only 12 patients in the trajectory planning group. Because all the symptomatic ICH patients had not ever received this recording, we could not determine whether MER had an effect on the difference in the occurrence of ICH between the two groups.

Planning was performed to determine entry points for a safe electrode trajectory that avoided blood vessels, sulci, and ventricles. Careful preoperative planning to avoid all potential vessels crossing the proposed trajectory may be crucial to avoid hemorrhage [1]. As a general rule, a minimum distance of 2 mm from vessel-like structures and the ventricular wall is enforced for the trajectory [2,21]. In our series, most trajectories were maintained at least 4 mm away from these important structures. Our institution has dedicated extensive efforts to reducing the likelihood of immediate and delayed ICH after DBS surgery. Care is taken to avoid the superficial and deep sulci, as well as the ventricles. Moreover, to prevent vessel injury from the trajectory, some special MRI scans were performed to better visualize the vessels, and the surgical plan was adjusted to keep the electrode trajectory away from any vessels [2].

For the selection of the image fusion method to plan the trajectory, stereotactic contrast-enhanced T1W imaging has been used to visualize vessels to prevent ICH. SWI, which is superior to T1W imaging, is also used for DBS trajectory planning [22]. Among the potential sources of ICH are mesencephalic veins at the endpoints of the electrode paths, which can cause fatal hemorrhage, and these veins can be visualized reliably with SWI. SWI allows the improved visualization of smaller, deep vessels, whereas T1W-Gd adequately detects larger superficial vessels. However, increased detection by SWI does not appear to significantly benefit trajectory planning [23]. Trajectories planned with T1W structural MRI, susceptibility-weighted venography, and time-of-flight angiography (T1W-SWI-TOF) do not suffer from missing vascular information or imprecise data set registration. The proposed protocol had minimal effects on the imaging and surgical workflows and facilitated vessel avoidance, thus providing a safe, cost-effective alternative to the injection of gadolinium contrast medium [24]. Careful application of neuronavigation using gadolinium-enhanced MRI of the brain has been advocated for the selection of the safest electrode trajectories [25]. Targeting based on merging computed tomography angiography (CTA) and T1W/T2W MRI has been shown to further increase the safety of lead placement and to reduce the risk of bleeding-related sequelae [3]. The combination of CTA and contrast-enhanced MRI also increases the likelihood of detecting vascular conflicts along the trajectory [26]. SWI, TOF, and T1W-Gd were performed to ensure clear, intense signals from the vessels to prevent vascular injury in the brain parenchyma [2]. Although double-dose enhanced MRI has been advocated for planning, Tonge M et al. reported no difference in the risk of ICH between patients who underwent surgery with single-dose or double-dose enhanced MRI for planning [21]. In our study, multiple imaging modalities, including 3D BRAVO imaging, SWI, TOF MRA, double-dose gadolinium-enhanced MRI, stereotactic frame-placed MRI, and fused images, were applied for trajectory planning, which resulted in effective clinical outcomes.

Cranial imaging examination showed that most intracranial bleeding surrounded electrodes as the distribution center, while some bleeding was more remote, with normal brain tissue between the lead and hemorrhage owing to cerebral infarction caused by local arteriovenous injury or coagulation of a cortical vein [12]. Most intracranial bleeding after DBS occurred near small arteries or veins due to compression by a microelectrode tip or rigid guide cannula along the trajectory of the electrode [27]. Electrode implantation without trajectory planning may easily lead to other complications in addition to ICH. In our study, one patient’s 1-year postoperative CT scan showed that one lead was displaced from its original position and had coiled in the lateral ventricle. The electrode trajectory had not been planned, and the electrode was speculated to pass through the lateral ventricle and then become dislodged by cerebrospinal fluid (CSF) after surgery. Apart from displacement of the lead from the target after surgery, passing a cannula through the lateral ventricle may lead to CSF leakage, which can easily cause inaccurate contact placement during surgery.

Reaching deep targets safely is difficult because surgeons must plan trajectories that avoid critical structures and reach targets at specific angles [28]. Improvement of clinical efficacy sometimes demands additional target penetration when planning a trajectory. Alignment of the Vim and posterior subthalamic area (PSA) for DBS is feasible; thus, a parietal trajectory facilitates the positioning of electrode contacts in both areas, yielding better control of tremor symptoms in PD and essential tremor (ET) [29,30]. Richieri R et al. reported that an electrode trajectory was calculated to pass through the anterior part of the middle frontal gyrus (Brodmann area 9/46) and the ALIC before ending in the ventral striatum (namely, the NAc) [31]. In our ALIC/NAc cases, since the ALIC is located dorsolateral to the NAc, we first used the “path” function in SurgiPlan to set the target to the NAc and then adjusted the trajectory to the shape of the dorsal anterior part of the ALIC. Finally, we plotted a trajectory through the prefrontal cortex as the entry point and ensured that the whole path avoided blood vessels, sulci, and ventricles. Therefore, the ring and arc angle values were significantly different from those of the other three targets, and their trajectories were directed more anteriorly and laterally. In our other cases, the locations and coordinates of the targets mainly determined the angle of the trajectory, and avoidance of important structures played only a relatively minor role. Because the STN and Vim had similar X coordinates, no significant difference in the arc angle was observed between the STN and Vim groups. Furthermore, the standard deviation of the ring angle value depicted in Figure 2 was obviously greater than that of the arc angle value, possibly illustrating that we adjusted and widened the ring angle more frequently than the arc angle when endeavoring to avoid the vasculature. Both the ring and arc angles for each given target shared their own relatively constant ranges by which the surgeons could easily identify frame displacement and other potentially fatal mistakes before inserting the cannula into the brain.

In order to define the “ring and arc zone of danger” for lead placement in the STN, the left ring and arc angle values of trajectories between the ICH cases and no ICH cases were compared. We found that the ICH cases had a trend of smaller left ring and arc values, possibly indicating that the trajectory with a relatively small angle with the axial and sagittal planes increased the risk of ICH, and the “ring zone of danger” was approximately between 60° and 66°. We plan to wait for more samples to confirm this conclusion.

The results for each target’s simulated trajectory allow us to recognize the specific deep brain structures in the path of a lead and increase our awareness of some correlative neurological side effects, which probably result from a small hematoma, infarction, edema, and microstructural injury along the lead in the basal ganglia area. Downes AE et al. reported that five patients developed acute-onset lethargy, dysarthria, and hemibody weakness intraoperatively due to ischemic stroke during GPi DBS [32]. The case shown in Figure 5 is a typical example of a lesion of the left PLIC in the trajectory for STN DBS. Additionally, the data on trajectory angles can guide us in adjusting the arc and ring according to the given nucleus if trajectory planning cannot be completed preoperatively, as large vessels are usually avoided in a relatively constant trajectory.

## 5. Conclusions

Trajectory planning based on MRI image fusion is a safe technique for electrode placement and has reduced the incidence of ICH after DBS surgery in our center. Preoperative brain images, including 3D BRAVO, SWI, TOF MRA, and T1W-Gd images, are helpful for the detection of the deep sulci and vasculature. Depending on the locations of different nuclei, if the entry point is set to a common area on the scalp, the arc and ring angles for the trajectories are significantly different, and the anatomic structures through which the leads pass at the levels of the lateral ventricle and the internal capsule also differ; namely, a trajectory’s arc and ring angles and traversed anatomic structures are distinctive for different nuclei. Considering the detailed analysis, we conclude that the electrode for each given target has its own relatively constant trajectory.

This study has led us to advocate for the following brief protocol: (1) the lateral beam is maintained parallel to the AC-PC line when placing the frame; (2) the burr hole site is roughly marked 0.0~1.5 cm anterior to the coronal suture and 3.5~4.5 cm lateral to the midline; (3) the stereotactic angle is initially set based on the target and its “tailored” values; and (4) the trajectory is meticulously adjusted according to MRI images, specifically, the 3D BRAVO imaging can help the surgeon to avoid sulci, and the SWI and enhanced MRI can help the surgeon to avoid superficial and deep vessels.

## 6. Limitations

One of this study’s limitations is that the admitted patients were not randomly divided into the trajectory planning group and the non-trajectory planning group. Trajectory planning based on co-registered brain images was not performed in random order, and only the most recent patients were enrolled due to technical progress in DBS in our center. Speculating that the first patients tended to have a higher incidence of ICH is unreasonable. The learning curve effect should be considered minimal because the surgical procedure and perioperative therapy were carried out by the same experienced surgeon and his team, who had completed DBS surgery for approximately 500 patients from 1999 to 2008. Moreover, this study focuses only on hemorrhage as a typical important complication, but other vascular-related cerebral injuries, including edema and non-hemorrhagic cerebral infarction, will be investigated in further research. Additionally, we did not perform the postoperative CT thin-layer scanning routinely; hence, the fusion of preoperative MRI scans and postoperative CT scans could not be realized in the same patients, including the three symptomatic ICH cases from the trajectory planning group, in our study. Therefore, it is difficult to identify the specific zone of danger for ICH. Although several papers have been published on the risk of ICH during DBS procedures and have proposed preoperative planning [9,13], this study retrospectively analyzed a larger single series of trajectory planning for DBS surgery and symptomatic ICH after DBS surgery in a historical cohort, not only to address the essential role of correct electrode placement but also to provide some more informative and useful results.

## Figures and Tables

**Figure 1 brainsci-12-00967-f001:**
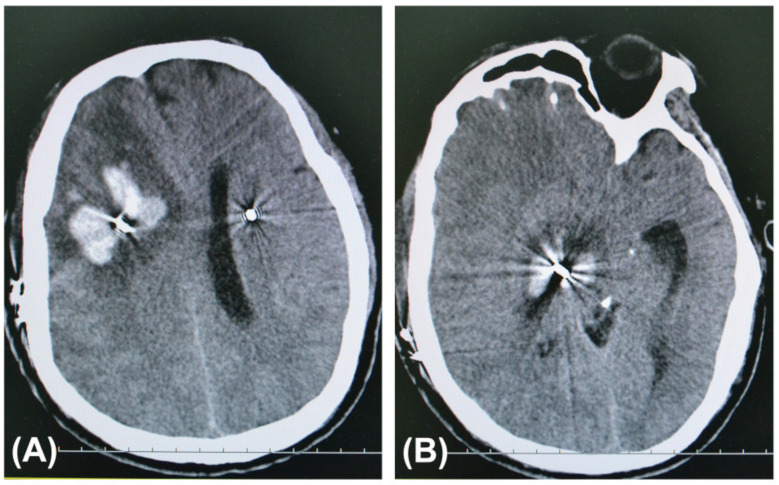
Axial computed tomography (CT) slices at the hemorrhage sites taken after the health of a 53-year-old man with PD in the non-trajectory planning group deteriorated on the 6th postoperative day. He had suffered from a headache for a few days before falling into a coma. His hematomas were along the trajectory of the definitive electrode in the right frontal lobe (**A**), basal ganglia, and midbrain (**B**).

**Figure 2 brainsci-12-00967-f002:**
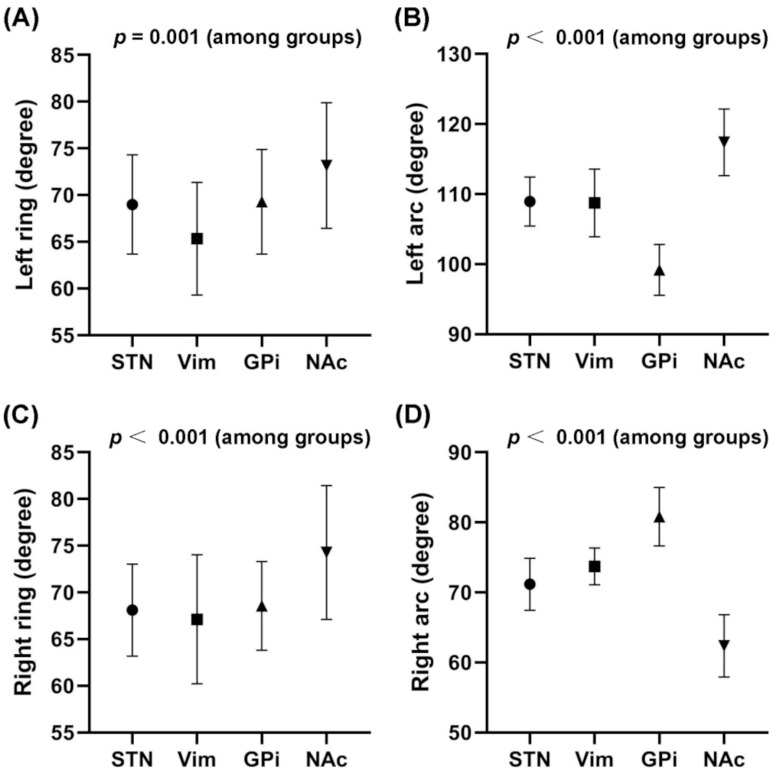
Ring and arc angle values of trajectories (mean ± SD). (**A**) Left ring: between the STN and GPi groups, *p* = 0.795; between the STN and Vim groups, *p* = 0.098; between the Vim and GPi groups, *p* = 0.093; between the GPi and NAc/ALIC groups, *p* = 0.007; between the Vim and NAc/ALIC groups, *p* = 0.001; and between the STN and NAc/ALIC groups, *p* < 0.001. (**B**) Left arc: between the STN and Vim groups, *p* = 0.903; and between any other two groups, *p* < 0.001. (**C**) Right ring: between the STN and GPi groups, *p* = 0.678; between the STN and Vim groups, *p* = 0.637; between the Vim and GPi groups, *p* = 0.521; between the Vim and NAc/ALIC groups, *p* = 0.002; between the STN and NAc/ALIC groups, *p* < 0.001; and between the GPi and NAc/ALIC groups, *p* < 0.001. (**D**) Right arc: between the STN and Vim groups, *p* = 0.097; and between any other two groups, *p* < 0.001.

**Figure 3 brainsci-12-00967-f003:**
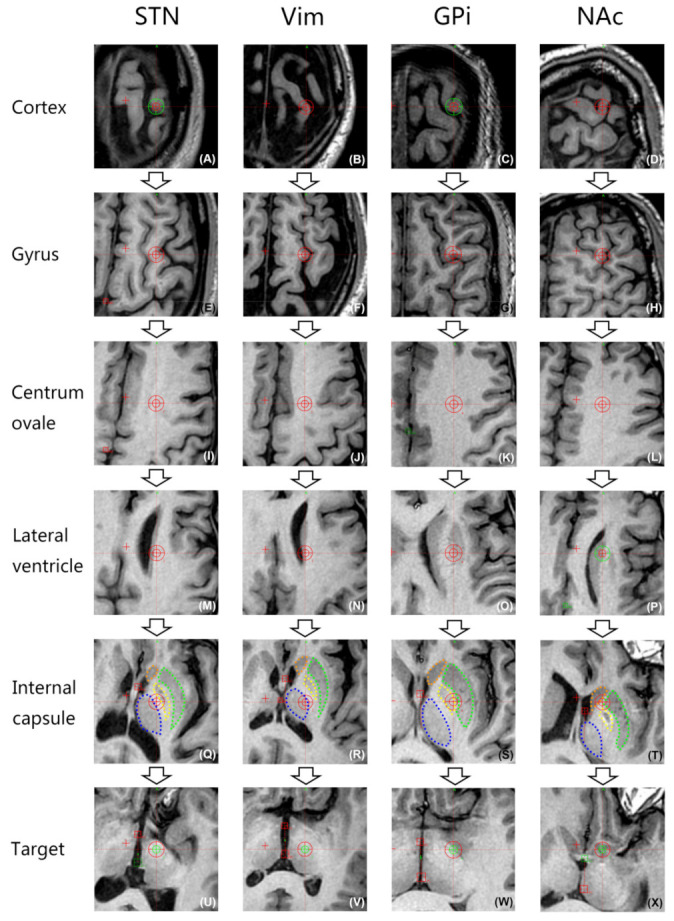
“Probe’s-eye” 3D BRAVO MRI images of different slices along the whole trajectory for main targets. (**A**–**D**) The trajectory enters the cortex. (**E**–**H**) The trajectory avoids the bilateral sulci. (**I**–**L**) The trajectory passes through the centrum ovale. (**M**–**P**) Upper part of the lateral ventricle; at this level, the planned trajectories for the STN and Vim both pass posterolaterally to the lateral ventricles close to their lateral walls (**M**,**N**); the trajectory for the GPi passes laterally to the lateral ventricle far from its lateral wall (**O**); the trajectory for the NAc passes anterolaterally to the lateral ventricle near its lateral wall (**P**). (**Q**–**T**) Internal capsule; at this level, the planned trajectory for the STN passes through the posterior limb of the internal capsule (PLIC) (**Q**); the trajectory for the Vim passes medially to the PLIC and enters the thalamus from its dorsolateral part (**R**); the trajectory for the GPi passes laterally to the PLIC and enters the globus pallidus from its dorsal part (**S**); the trajectory for the NAc passes through the ALIC (**T**); blue dotted line: thalamus; green dotted line: globus pallidus externa (GPe); yellow dotted line: GPi; orange dotted line: head of the caudate nucleus. (**U**–**X**) Locations of the targets.

**Figure 4 brainsci-12-00967-f004:**
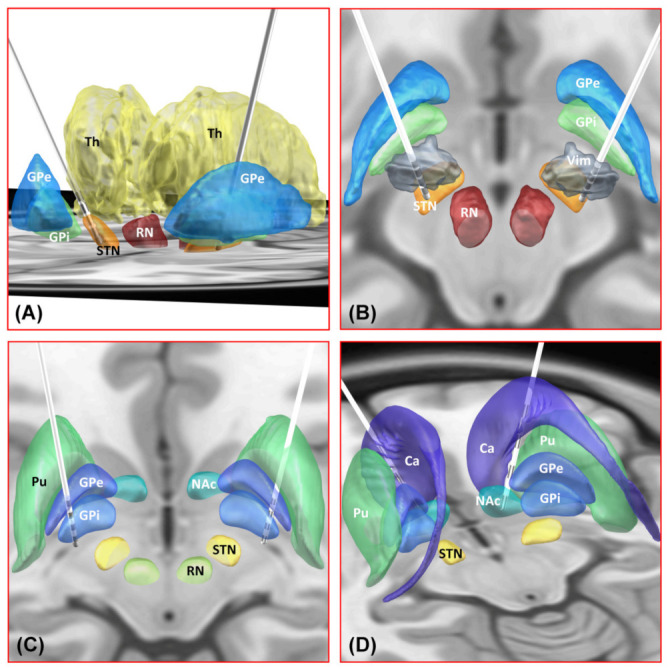
Deep brain stimulation (DBS) electrode reconstructions and computer simulations based on preoperative MRI imaging and postoperative CT imaging using the Lead-DBS toolbox (www.lead-dbs.org, accessed on 5 May 2022). These illustrations show the spatial relationship between bilateral leads for the STN (**A**), Vim (**B**), GPi (**C**), NAc (**D**), and their nearby structures. Ca: caudate nucleus; GPi: globus pallidus internus; GPe: globus pallidus externus; NAc: nucleus accumbens; RN: red nucleus; STN: subthalamic nucleus; Th: thalamus; Pu: putamen; Vim: ventralis intermedius nucleus of the thalamus.

**Figure 5 brainsci-12-00967-f005:**
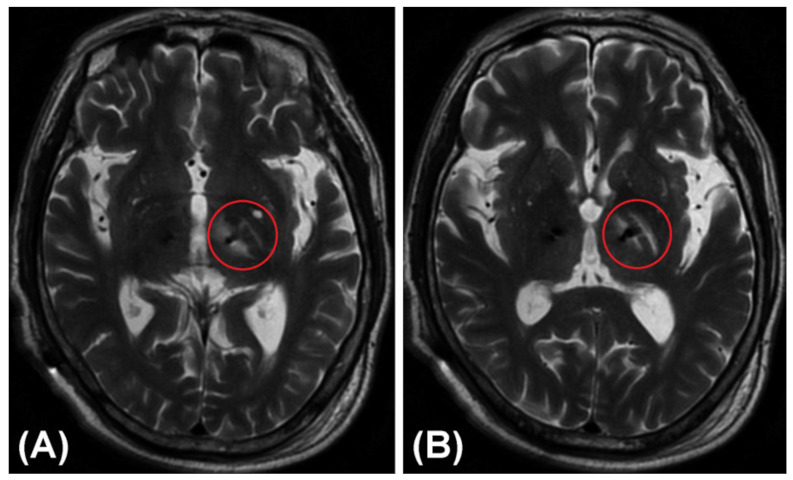
A rare non-hemorrhagic edematous lesion or infarction around the trajectory in a T2-weighted MRI image. An electrode placed for the left STN through the PLIC led to local edema of the PLIC and adjacent white matter (i.e., the medial medullary lamina), which appeared similar to a “hamburger”, together with the relatively normal GPi. (**A**) Lower layer. (**B**) Upper layer.

**Figure 6 brainsci-12-00967-f006:**
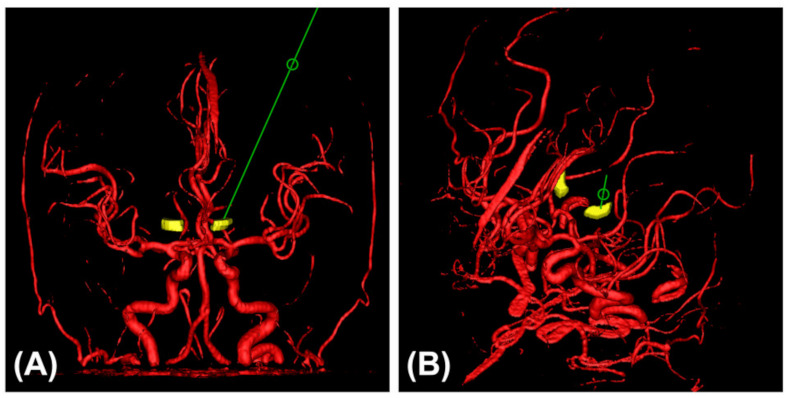
A simulated trajectory (green line) and the cerebral arteries (red vessels) on 3D MRA (yellow nuclei: the STN). (**A**) Frontal view image. (**B**) Approximate path’s-eye image. The trajectory was safe for lead placement in the STN.

**Figure 7 brainsci-12-00967-f007:**
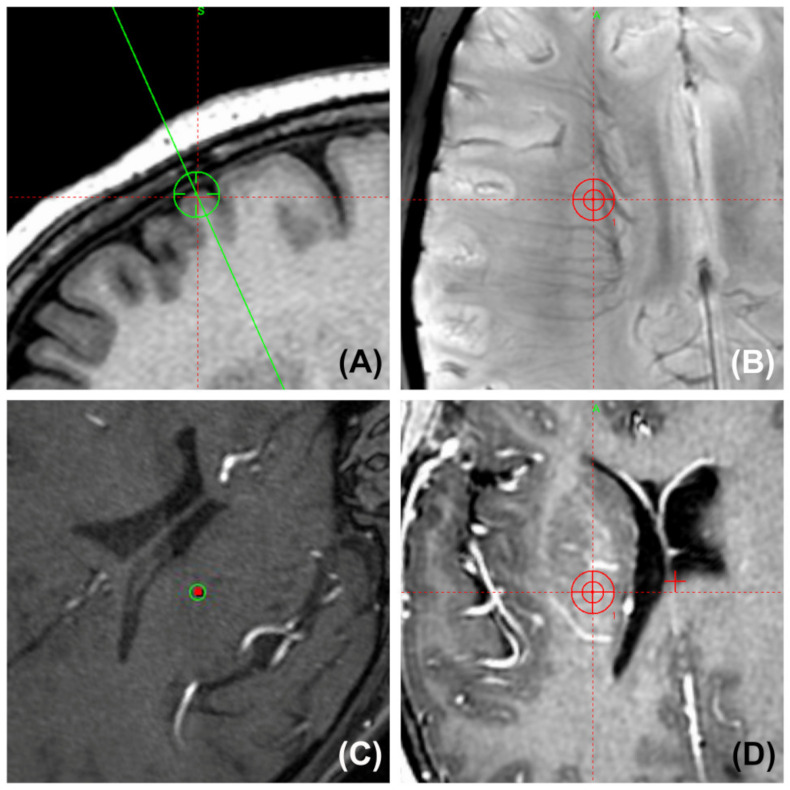
Surgical planning snapshot images show the position of the planned trajectory and its surrounding vessels (two-dimensional probe’s-eye trajectory visualization by SWI, TOF MRA, and T1W-Gd). (**A**) The entry point is always placed anterior to the coronal suture to avoid injuring the motor region. (**B**) Preventing the guide tube from puncturing small and deep vessels in the corona radiata by SWI. (**C**) Confirming that the trajectory was kept far from the arteries in the lateral fissure by TOF MRA. (**D**) Gadolinium-enhanced MRI shows that the trajectory was kept at a distance from the surrounding arteries and veins in the basal ganglia area.

**Table 1 brainsci-12-00967-t001:** Clinical characteristics of patients receiving deep brain stimulator lead implantation surgery.

	No Trajectory Planning	Trajectory Planning	*p*-Value
**Number of male patients**	**211** (53.3%)	**340** (54.6%)	0.687 ^※^
**Diagnosed with hypertension**	**42** (10.6%)	**109** (17.5%)	0.003 ^※^
**Age at surgery ***	**59** (49, 66) ***	**64** (54, 68) ***	0.000 ^※※^
**Local anesthesia**	**289** (73.0%)	**445** (71.4%)	0.591 ^※^
**Number of enrolled patients**	**396**	**623**	0.647 ^※※※^
Parkinson’s disease	318 (80.3%)	519 (83.3%)	
Essential tremor	5 (1.3%)	12 (1.9%)	
Dystonia	47 (11.9%)	53 (8.5%)	
Tourette’s syndrome	4 (1.0%)	6 (1.0%)	
Addiction	13 (3.3%)	17 (2.7%)	
Other psychiatric diseases	7 (1.8%)	11 (1.8%)	
Huntington’s disease	2 (0.5%)	4 (0.6%)	
Chronic pain	0 (0.0%)	1 (0.2%)	
**Number of implanted leads** **	**691**	**1237**	0.402 ^※※※※^
STN	557 (80.6%)	1032 (83.4%)	
Vim	22 (3.2%)	31 (2.5%)	
GPi	70 (10.1%)	117 (9.5%)	
NAc/ALIC	40 (5.8%)	56 (4.5%)	
CM-Pf	2 (0.3%)	0 (0.0%)	
VPL/VPM	0 (0.0%)	1 (0.1%)	

STN: subthalamic nucleus; Vim: ventralis intermedius nucleus of the thalamus; GPi: globus pallidus internus; NAc: nucleus accumbens; ALIC: anterior limb of the internal capsule; CM-Pf: centromedianus–parafascicularis complex; VPL: ventral posterolateral nucleus of the thalamus; VPM: ventral posteromedial nucleus of the thalamus. * 33 patients with Parkinson’s disease (PD) underwent electrode implantation at two different stages. ** Each implanted electrode was intended for only one target. *** The descriptive statistics for each group are presented as follows: the median (Q1, Q3). ^※^ Pearson’s chi-squared test was used. ^※※^ The Mann–Whitney U test was used. ^※※※^ Fisher’s exact test was used for PD, essential tremor, dystonia, Tourette’s syndrome, addiction (drug or alcohol), other psychiatric diseases (e.g., obsessive–compulsive disorder, treatment-resistant depression, anorexia nervosa, and refractory anxiety disorder), and Huntington’s disease and chronic pain, which were combined into one category. ^※※※※^ Pearson’s chi-squared test was used; CM-Pf and VPL/VPM were combined into one category.

**Table 2 brainsci-12-00967-t002:** Stereotactic systems and surgery planning software in the two groups.

Surgery Planning Systems/Headframe	No Trajectory Planning	Trajectory Planning	Total Number
**Total number** *	**396**	**623**	**1019**
Leksell SurgiPlan	279	456	735
CRW StereoCalc	117	141	258
Medtronic Framelink	0	26	26
***p*-Value**			0.000 **
Leksell	279	459	738
CRW	117	164	281
***p*-Value**			0.262 **

* 33 patients with PD underwent electrode implantation at two different stages. ** Pearson’s chi-squared test was used.

**Table 3 brainsci-12-00967-t003:** ICH after DBS surgery in the two groups.

ICH	No Trajectory Planning		Trajectory Planning		Total Number	
	No. of Patients	No. of Leads	No. of Patients	No. of Leads	Patients	Leads
**Symptomatic ICH**	10 *	10 **	3 *	3 **	**13**	**13**
**Without symptomatic ICH**	386 *	681 **	620 *	1234 **	**1006**	**1915**
**Total number**	**396**	**691**	**623**	**1237**	**1019**	**1928**
***p*-value**					0.005 *	0.003 **

ICH: intracranial hemorrhage. * Pearson’s chi-squared test was used. ** Fisher’s exact test was used.

**Table 4 brainsci-12-00967-t004:** Fusion of various MRI sequences.

Fused MRI Images	Number (%)
A + B	46 (7.4%)
A + B + C	219 (35.2%)
A + B + D	20 (3.2%)
A + B + E	278 (44.6%)
A + B + C + E	54 (8.7%)
A + B + C + D + E	6 (1.0%)
**Total number**	**623**

A: T1- or T2-weighted magnetic resonance imaging (MRI) taken after the frame was placed; B: three-dimensional brain volume (3D BRAVO) imaging; C: susceptibility-weighted imaging (SWI); D: time-of-flight magnetic resonance angiography (TOF MRA); E: T1-weighted gadolinium-enhanced MRI (T1W-Gd).

## Data Availability

The data that support the findings of this study are available from the corresponding author, upon reasonable request.

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
