# Peer review of "Optimized Deep Brain Stimulation Surgery to Avoid Vascular Damage: A Single-Center Retrospective Analysis of Path Planning for Various Deep Targets by MRI Image Fusion"

_brainsci, 2022, doi:10.3390/brainsci12080967_

Round 1

Reviewer 1 Report

The paper is generally well-written, fluent and clear in several of its part. The methodology is robust and interesting. The discussion section is extended and precise.

The following few minor issues remains:

2.3: Please provide a summary table for the technical details of the MRI scans. Also, use T_{1}, using the subscripts. I think that the authors should also provide more dtails related to the RF coils used in the study. 

2.6: The statistical analysis is appropriate.

Figure 2: the p-value of the sub-figure B-D is 0? Please provide additional details. 

Figure 7: please remove the old caption  (lines 290-295)

Author Response

Dear Editors and Reviewers:

Thank you for your letter and for the reviewers’ comments about our manuscript titled “Optimized Deep Brain Stimulation Surgery to Avoid Vascular Damage: A Single-Center Retrospective Analysis of Path Planning for Various Deep Targets by MRI Image Fusion”(brainsci-1736540). All the comments are very valuable, helped us revise and improve our manuscript, and guided our studies. We have carefully studied these comments and have made corrections, which we hope will meet with approval. The revised portions are highlighted in red in the revised manuscript. The main corrections in the paper and the responses to the reviewer’s comments are as follows:

Response to Reviewer 1 Comments

Point 1: Please provide a summary table for the technical details of the MRI scans. Also, use T_{1}, using the subscripts. I think that the authors should also provide more dtails related to the RF coils used in the study. 

Response 1: Thank you for your valuable suggestions.I have provided a summary table for the technical details of the MRI scans. Please find it in the Supplementary Materials. An orthogonal RF coil was used for head after stereotactic frame placement. As for “use T_{1}, using the subscripts”, I have changed“T1- or T2-weighted” to “T_1 or T_2-weighted” in the text.

Point 2: 2.6: The statistical analysis is appropriate. Figure 2: the p-value of the sub-figure B-D is 0? Please provide additional details.  

Response 2: Thank you for the revision suggestion. I have replaced“P=0.000” with “P<0.001”in the text.

Point 3: Figure 7: please remove the old caption (lines 290-295)  

Response 3: Thank you for the revision suggestion. I have removed the repeated caption.

Reviewer 2 Report

The authors present a large case series of patients undergoing implantation of DBS electrodes at a single center and  have to be congratulated for their work on analyzing this significant amount of patients. 

The authors claim higher rates of ICH if implantation trajectories are not intentionally planned to avoid vessels and ventricles comparing to an older  cohort after changing the surgical/planning regime.

Nevertheless many important aspects remain unclear or cannot be reported due to the retrospective nature of the analysis. 

- Only symptomitic ICH can be compared as no regular imaging was performed in a significant amount of patients, how many?

- Further details on the surgical procedure need to be shown, was there microelectrode recording? How many surgeries were performed awake?

- One group consisted of patients undergoing lead implantation based on a relatively standardized burr hole defining the trajectory by the center of arc principle. How were the target points defined?  ACPC coordinates or direct  targeting?

- A retrospective analysis of trajectories in patients that had symptomatic bleedings via fusion of preoperative MRI scans and postoperative CT scans could be of interest to identify actual zones of danger?

- The rational to avoid sulci, ventricles and blood vessels can be regarded as common knowledge in stereotactic procedures and might only be challenged in structures studies. 

- The analysis of, and description of significant differences of  trajectory values in the arc space does not add any relevant information by itself. Is there an attribution of specific arc setting with symptomatic  bleeding to define an "arc zone of danger"?

Author Response

Dear Editors and Reviewers:

Thank you for your letter and for the reviewers’ comments about our manuscript titled “Optimized Deep Brain Stimulation Surgery to Avoid Vascular Damage: A Single-Center Retrospective Analysis of Path Planning for Various Deep Targets by MRI Image Fusion”(brainsci-1736540). All the comments are very valuable, helped us revise and improve our manuscript, and guided our studies. We have carefully studied these comments and have made corrections, which we hope will meet with approval. The revised portions are highlighted in red in the revised manuscript. The main corrections in the paper and the responses to the reviewer’s comments are as follows:

Response to Reviewer 2 Comments

Point 1: Only symptomitic ICH can be compared as no regular imaging was performed in a significant amount of patients, how many? 

Response 1: Thank you for your valuable question. No regular postoperative imaging was performed in about half of the 396 patients in no trajectory planning group. Their cranial CT scan was not performed on the first day after operation.

Point 2: Further details on the surgical procedure need to be shown, was there microelectrode recording? How many surgeries were performed awake?  

Response 2: Thank you for your valuable question. The microelectrode recording (MER) was seldom used in our study, and all the symptomatic ICH patients had not ever received this recording. We reviewed our case series carefully and found the microelectrode recording was utilized to refine targeting in 31 patients in no trajectory planning group, and was utilized in only 12 patients in trajectory planning group. Therefore, we can not understand whether MER has an effect on the difference of occurrence of ICH between the two groups. Moreover, the majority of PD patients required local anesthesia for electrode implantation, during which they were carefully observed for curative effects and side effects of macrostimulation. The number proportion of DBS surgeries performed awake was not different between the two groups.The results were added in Table 1.

Point 3: One group consisted of patients undergoing lead implantation based on a relatively standardized burr hole defining the trajectory by the center of arc principle. How were the target points defined?  ACPC coordinates or direct targeting?  

Response 3: Thank you for your valuable questions. Both in the “hole-based trajectory” group and “imaging-based trajectory” group, the target points were defined by the neurosurgeon in the surgery slanning softwares according to MRI images which had been imported in advance. All the nuclei were targeted directly, AC-PC line helped us define the coordinates of nuclei through the relatively constant spatial relationship between them.

Point 4: A retrospective analysis of trajectories in patients that had symptomatic bleedings via fusion of preoperative MRI scans and postoperative CT scans could be of interest to identify actual zones of danger? 

Response 4: Thank you for raising this important issue. This comment directs our next in-depth research. In our previously published article (PMID: 27760466), we described the locations of hematomas in intracerebral hemorrhage (ICH) cases from the no trajectory planning group. The hemorrhages mainly occurred in the cerebral lobes and basal ganglia, and they were around or far away from the electrodes due to their different features. Furthermore, because of the diversity of targets involved in this study, it is difficult to identify the actual zones of danger for ICH.

Point 5: The rational to avoid sulci, ventricles and blood vessels can be regarded as common knowledge in stereotactic procedures and might only be challenged in structures studies. 

Response 5: Thank you for the insightful comments. Frankly, an effective method to reduce ICH caused by lead implantation surgery is to plan the trajectory of implanted electrodes to avoid ventricles, sulci and blood vessels in the brain. However, the reliable case-control studies and convincing evidence are needed to further verify this idea. Additionally, our study provided the trajectory angles and brain structures intersected for lead placements at the frequently used nuclei for the first time, which could help us avoid the adverse events.

Point 6: The analysis of, and description of significant differences of  trajectory values in the arc space does not add any relevant information by itself. Is there an attribution of specific arc setting with symptomatic  bleeding to define an "arc zone of danger"? 

Response 6: Thank you for raising this important issue. We started to record arc values in the trajectory planning group, and the incidence of ICH in this group was very low with only 3 cases; therefore, it was difficult to define the "arc zone of danger" due to the small sample size. However, based on our experience in this topic, the arc value with a small angle with the midline increases the risk of entering the lateral ventricle. The arc value with a large angle is easy to make the temporal muscle damaged with more bleeding amount and difficult operation. Moreover, in order to ensure the curative effect, some nuclei have a certain range when setting the arc value to ensure that the most contacts of leads are located in the nuclei. For example, the arc value range of STN is 15 °- 25 °.

Reviewer 3 Report

The topic is extremely important in clinical treatment, and breakthroughs in research are needed to improve the approach with less risk to the patient. I just have a few suggestions.

The indication of whose Ethics Committee approved the study in the methods section, as well as the completeness of the permission number was not cited.

Another recommendation is to move table 1 from the methods section to the results section (item 3.1), because this item contains a description of the analysis contained in table 1, which describes clinical data statistical analysis.

The article contains a huge number of acronyms, some of which are not defined in the first reference.

The information described in items 5 and 6 of the manuscript is inserted in the wrong topic. I suggest transferring the paragraph from the manuscript conclusion section to the end of the discussion section, and transferring the text of the patents item to the conclusion section of the manuscript.

Author Response

Dear Editors and Reviewers:

Thank you for your letter and for the reviewers’ comments about our manuscript titled “Optimized Deep Brain Stimulation Surgery to Avoid Vascular Damage: A Single-Center Retrospective Analysis of Path Planning for Various Deep Targets by MRI Image Fusion”(brainsci-1736540). All the comments are very valuable, helped us revise and improve our manuscript, and guided our studies. We have carefully studied these comments and have made corrections, which we hope will meet with approval. The revised portions are highlighted in red in the revised manuscript. The main corrections in the paper and the responses to the reviewer’s comments are as follows:

Response to Reviewer 3 Comments

Point 1: The indication of whose Ethics Committee approved the study in the methods section, as well as the completeness of the permission number was not cited. 

Response 1: Thank you for your helpful suggestion. This study was approved by the Institutional Review Board of Tangdu Hospital, Fourth Military Medical University on Jun 22nd, 2017. The ethical code number is TDLL-201706-28. The information has been inserted in the methods section.

Point 2: Another recommendation is to move table 1 from the methods section to the results section (item 3.1), because this item contains a description of the analysis contained in table 1, which describes clinical data statistical analysis. 

Response 2: Thank you for your valuable suggestion. I have moved table 1 from the methods section to the results section (item 3.1).

Point 3: The article contains a huge number of acronyms, some of which are not defined in the first reference. 

Response 3: Thank you for your helpful suggestion.I have defined all the acronyms in the first reference.

Point 4: The information described in items 5 and 6 of the manuscript is inserted in the wrong topic. I suggest transferring the paragraph from the manuscript conclusion section to the end of the discussion section, and transferring the text of the patents item to the conclusion section of the manuscript. 

Response 4: Thank you for the revision suggestions. I have made modifications according to your suggestions. 

Round 2

Reviewer 2 Report

After the first review, I do not see substantial changes in this manuscript according to my remarks. Therefore, I cannot  support this manuscript for publication in the present form.

Author Response

Dear Reviewer:

Thank you for your comments about our manuscript titled “Optimized Deep Brain Stimulation Surgery to Avoid Vascular Damage: A Single-Center Retrospective Analysis of Path Planning for Various Deep Targets by MRI Image Fusion”(brainsci-1736540). All the comments are very valuable, helped us revise and improve our manuscript, and guided our studies. After the first round of revision, we carefully studied all the comments from reviewer 2 and made major corrections and improvements, especially in terms of the details on surgical procedures and studies on "ring and arc zone of danger", which we hope will meet with your approval. The revised portions are highlighted in red in the revised manuscript. The main corrections in the paper and the responses to your comments are as follows:

Point 1: Only symptomitic ICH can be compared as no regular imaging was performed in a significant amount of patients, how many? 

Response 1: Thank you for your valuable question. No regular postoperative imaging was performed in 186 patients in no trajectory planning group. Their cranial CT scan was not performed on the first day after operation. The result has been added in “3.Results 3.3. Comparison of ICH After DBS Surgery Between the Two Groups”.

Point 2: Further details on the surgical procedure need to be shown, was there microelectrode recording? How many surgeries were performed awake?  

Response 2: Thank you for your valuable question. The microelectrode recording (MER) was seldom used in our study, and all the symptomatic ICH patients had not ever received this recording. We reviewed our case series carefully and found the microelectrode recording was utilized to refine targeting in 31 patients in no trajectory planning group, and was utilized in only 12 patients in trajectory planning group. Therefore, we can not understand whether MER has an effect on the difference of occurrence of ICH between the two groups. These sentences were added in “4. Discussion”. Moreover, the majority of PD patients required local anesthesia for electrode implantation, during which they were carefully observed for curative effects and side effects of macrostimulation. The number proportion of DBS surgeries performed awake was not significantly different between the two groups.The results were added in Table 1.

Point 3: One group consisted of patients undergoing lead implantation based on a relatively standardized burr hole defining the trajectory by the center of arc principle. How were the target points defined?  ACPC coordinates or direct targeting?  

Response 3: Thank you for your valuable questions. Both in the “hole-based trajectory” and “imaging-based trajectory” surgical procedure, the trajectory was defined by the center of arc principle, and the target points, namely the center of arc were defined by the neurosurgeon in the surgery planning softwares based on the MRI images which had been imported in advance. All the nuclei were targeted directly, AC-PC line only helped us confirm the coordinates of nuclei through the relatively constant spatial relationship between them. The differences between these two procedures were that the relatively standardized burr hole and the target point determined the path in the former procedure, and the well-planned trajectory and the target point determined the burr hole in the latter procedure. These statements have been inserted in “2. Materials and Methods  2.3. Surgical Procedure and Trajectory Planning” in order to describe the surgical procedures profoundly.

Point 4: A retrospective analysis of trajectories in patients that had symptomatic bleedings via fusion of preoperative MRI scans and postoperative CT scans could be of interest to identify actual zones of danger? 

Response 4: Thank you for raising this important issue. Actually, we did not perform the postoperative CT thin-layer scanning routinely, hence the fusion of preoperative MRI scans and postoperative CT scans can not be realized in the some patients including the three ICH cases from the trajectory planning group in our study. This comment directs our next in-depth research. In terms of actual zones of danger, in our previously published article (PMID: 27760466), we described the locations of hematomas in intracerebral hemorrhage (ICH) cases from the no trajectory planning group. The hemorrhages mainly occurred in the cerebral lobes and basal ganglia, and they were around or far away from the electrodes due to their different features. Furthermore, because of the diversity of targets involved in this study, it is difficult to identify the specific zone of danger for ICH. These sentences have been stated in “6. Limitations”.

Point 5: The rational to avoid sulci, ventricles and blood vessels can be regarded as common knowledge in stereotactic procedures and might only be challenged in structures studies. 

Response 5: Thank you for the insightful comments. Frankly, an effective method to reduce ICH caused by lead implantation surgery is to plan the trajectory of implanted electrodes to avoid ventricles, sulci and blood vessels in the brain. However, the reliable case-control studies and convincing evidence are needed to further verify this idea. Additionally, our study provided the trajectory angles and brain structures intersected for lead placements at the frequently used nuclei for the first time, which could help us avoid the adverse events. At the end of the “6. Limitations”, we discussed as follows:“Although several papers have been published on the risk of ICH during DBS procedures and have proposed preoperative planning, this study retrospectively analyzed a larger single series of trajectory planning for DBS surgery and symptomatic ICH after DBS surgery in a historical cohort not only to address the essential role of correct electrode placement but also to provide some more informative and useful results.”

Point 6: The analysis of, and description of significant differences of  trajectory values in the arc space does not add any relevant information by itself. Is there an attribution of specific arc setting with symptomatic  bleeding to define an "arc zone of danger"? 

Response 6: Thank you for raising this important issue. We started to record ring and arc angle values which were automatically calculated by the Leksell SurgiPlan software in the trajectory planning group. The incidence of ICH in this group was very low, and all the three hemorrhage patients were diagnosed with PD and received the STN-DBS surgery. In spite of the small sample size of ICH patients, whose hemorrhages all occurred in the left hemispheres, we preliminarily compare the left ring and arc values between the ICH cases and no ICH cases in the trajectory planning group to define the "ring and arc zone of danger" for lead placement in the STN. The significant differences have not been found, but the hemorrhage cases had a trend of smaller left ring and arc values, possibly indicating that the trajectory with relatively small angle with the axial and sagittal planes increases the risk of ICH, and the "ring zone of danger" is approximately between 60°and 66°. We plan to wait for more samples to confirm this conclusion. Please find the results in “3. Results 3.5. Trajectory Angles Calculated by SurgiPlan for Four Targets”, Supplementary Materials (FIGURE S1), and 4. Discussion. Besides, based on our experience in this topic, the arc value with a small angle with the sagittal plane increases the risk of entering the lateral ventricle. The arc value with a large angle is easy to make the temporal muscle damaged with more bleeding amount and difficult operation. Moreover, in order to improve the therapeutic effect, some nuclei have a certain range when setting the arc value to ensure that the most contacts of leads are located in the nuclei. For example, the arc value range of STN is 15 °–25 °.